# Synthesis and Preclinical Evaluation of Radiolabeled [^103^Ru]BOLD-100

**DOI:** 10.3390/pharmaceutics15112626

**Published:** 2023-11-15

**Authors:** Barbara Happl, Marie Brandt, Theresa Balber, Katarína Benčurová, Zeynep Talip, Alexander Voegele, Petra Heffeter, Wolfgang Kandioller, Nicholas P. Van der Meulen, Markus Mitterhauser, Marcus Hacker, Bernhard K. Keppler, Thomas L. Mindt

**Affiliations:** 1Ludwig Boltzmann Institute Applied Diagnostics, General Hospital of Vienna, Waehringer Guertel 18-20, 1090 Vienna, Austriamarkus.mitterhauser@univie.ac.at (M.M.); 2Division of Nuclear Medicine, Department of Biomedical Imaging and Image Guided Therapy, Medical University of Vienna, Waehringer Guertel 18-20, 1090 Vienna, Austria; 3Institute of Inorganic Chemistry, Faculty of Chemistry, University of Vienna, Waehringer Strasse 42, 1090 Vienna, Austria; 4Joint Applied Medicinal Radiochemistry Facility, University of Vienna, Medical University of Vienna, 1090 Vienna, Austria; 5Center for Radiopharmaceutical Sciences, Paul Scherrer Institute, Forschungsstrasse 111, 5232 Villigen, Switzerland; 6Laboratory of Radiochemistry, Paul Scherrer Institute, Forschungsstrasse 111, 5232 Villigen, Switzerland; 7Institute of Cancer Research, Comprehensive Cancer Center, Medical University of Vienna, Borschkegasse 8A, 1090 Vienna, Austria; 8Research Cluster “Translational Cancer Therapy Research”, Waehringer Strasse 42, 1090 Vienna, Austria

**Keywords:** ruthenium-103, BOLD-100, anti-cancer metallodrugs, in vivo studies, pharmacokinetics

## Abstract

The first-in-class ruthenium-based chemotherapeutic agent BOLD-100 (formerly IT-139, NKP-1339, KP1339) is currently the subject of clinical evaluation for the treatment of gastric, pancreatic, colorectal and bile duct cancer. A radiolabeled version of the compound could present a helpful diagnostic tool. Thus, this study investigated the pharmacokinetics of BOLD-100 in more detail to facilitate the stratification of patients for the therapy. The synthesis of [^103^Ru]BOLD-100, radiolabeled with carrier added (c.a.) ruthenium-103, was established and the product was characterized by HPLC and UV/Vis spectroscopy. In order to compare the radiolabeled and non-radioactive versions of BOLD-100, both complexes were fully evaluated in vitro and in vivo. The cytotoxicity of the compounds was determined in two colon carcinoma cell lines (HCT116 and CT26) and biodistribution studies were performed in Balb/c mice bearing CT26 allografts over a time period of 72 h post injection (p.i.). We report herein preclinical cytotoxicity and pharmacokinetic data for BOLD-100, which were found to be identical to those of its radiolabeled analog [^103^Ru]BOLD-100.

## 1. Introduction

Ruthenium-based compounds are considered highly promising chemotherapeutic drug candidates [1,2]. KP1019 (indazolium *trans*-[tetrachloridobis(indazole)ruthenate(III)]) and NAMI-A (imidazolium *trans*-[tetrachlorido-(DMSO)(imidazole)ruthenate(III)] have already undergone clinical trials (Figure 1) [3,4]. NAMI-A was the first ruthenium-based anti-cancer drug that entered clinical trials; however, due to the lack of anti-tumor activity in the latest phase I/II clinical trial, further studies were not pursued [5]. Promising results for KP1019 were obtained in the phase I clinical trial, but its efficacy was jeopardized by its limited solubility requiring large injection volumes [6]. Consequently, the sodium salt of KP1019 with improved water solubility, termed BOLD-100 (formerly IT-139 or NKP-1339), was then used for further preclinical and clinical studies [7]. The exact mode of action of BOLD-100 is still under investigation. It has been shown that BOLD-100 binds to blood proteins such as albumin and transferrin [8]. The resulting drug–protein adduct is then believed to accumulate in tumor tissues, presumably mediated by the enhanced permeability and retention (EPR) effect [8,9]. BOLD-100 is a so-called prodrug that needs to be activated by the reduction of Ru(III) to Ru(II). In contrast to normal tissues, the electrochemical potential necessary for the reduction of the Ru(III) prodrug to an active Ru(II) species is found in the hypoxic environment of tumors [10]. Following this reduction step, the glucose-related protein (GRP) chaperone GRP78 in the endoplasmic reticulum (ER) has been reported as the likely intracellular target of the drug [11,12]. If cancer cells undergo ER stress, GRP78 is upregulated and activated, leading to cell survival and tumor growth [13]. BOLD-100 could interfere with the cell’s protein machinery and, as a consequence, apoptotic pathways are activated by the inhibition of GRP78, leading to apoptosis [7]. Alternatively, it has been shown that BOLD-100 inhibits DNA repair pathways and induces reactive oxygen species in breast cancer [14]. And, to clarify the biological pathways and elucidate the anti-cancer activity, cell line screening has been performed [15]. Despite the lack of knowledge about the exact mode of action, the phase I clinical study with BOLD-100 demonstrated a manageable safety profile at a maximum tolerated dose of 625 mg/m^2^, promising anti-tumor activity, lack of adverse effects, as well as the possibility of potentiating its activity in combination with other anti-cancer drugs [16]. Based on the promising results of the first phase I clinical trial, BOLD-100 is currently being tested in a phase Ib/IIa dose-escalation study in combination with FOLFOX for chemotherapy in patients with advanced solid tumors (NCT04421820). FOLFOX is a therapy regime which uses a combination of folinic acid, fluorouracil and oxaliplatin [17]. The combination of BOLD-100 with FOLFOX chemotherapy has been shown to be a well-tolerated treatment in a heavily pre-treated Stage IV biliary tract and gastric cancer study [18].

Because not all patients responded equally to the BOLD-100 therapy in clinical trials [16], a diagnostic imaging agent for the stratification of patients would be of great importance. Also, radiolabeled BOLD-100 could be useful in studying the pharmacokinetic profile of BOLD-100 and might provide insights into its mode of action. The aim is not to affect the biological behavior of BOLD-100 and therefore, radiolabeling without structural modification is desired, e.g., by application of a radioactive ruthenium isotope (Table 1). The medical use of radioactive ruthenium isotopes is not new. For example, ruthenium-106 implants have been used since 1964 for brachytherapy [19,20]. Single-photon emission computed tomography (SPECT) radionuclide ruthenium-97 (^97^Ru) appears to be the best option for molecular imaging with BOLD-100. It decays by electron capture, releasing 215.7 keV of γ-radiation, which is within the energy window of SPECT detectors (Table 1). However, the *β*^−^ emitter ruthenium-103 (^103^Ru) was utilized in this study as a model nuclide due to better availability. ^103^Ru can be produced via neutron irradiation of natural ruthenium (^nat^Ru) by the ^nat^Ru(n,γ)^103^Ru reaction [21]. In addition, its potential use for endoradiotherapy has been discussed in the literature [22].

Mainly in the 1970s and 1980s, a variety of ^97/103^Ru labeled compounds have been synthesized and evaluated preclinically [24]. For example, the chelator DTPA (diethylenetriaminepentaacetic acid) was radiolabeled with both radionuclides and [^97^Ru]Ru-DTPA was used for a cerebrospinal fluid imaging (cisternography) in dogs [25]. Also, preclinical evaluation was performed with radiolabeled ruthenocene derivatives [26] and transferrin [27]. Furthermore, [^103^Ru]RuCl_3_ scintigraphy was performed in patients with various types of malignant tumors [28]. More recently, Weiss et al. published the synthesis of a ^103^Ru-radiolabeled version of the organometallic Ru(II) complex RAPTA-C ([Ru(*η*^6^-*p*-cymene)Cl_2_(pta)], pta = 1,3,5-triaza-7-phospha-adamantane) as well as its preclinical evaluation in ovarian and colorectal carcinoma mouse models [29]. In addition, the complexation of ^97^Ru to a pyridine-2,6-dicarboxamide derivative was reported [30].

Herein, we report the isotopic radiolabeling of BOLD-100 with c.a. ^103^Ru and a side-by-side comparison of the obtained [^103^Ru]BOLD-100 with BOLD-100. After characterization of the compound, the cytotoxicity of [^103^Ru]BOLD-100 was determined in HCT116 and CT26 cells and its biodistribution was studied in tumor-bearing Balb/c mice. All experiments were also performed with the unlabeled compound BOLD-100 to evaluate a potentially enhanced therapeutic effect of the radiolabeled complex. The goal of this work was to verify that the radiolabeling of BOLD-100 with ruthenium radionuclides does not alter its biological properties and therefore could provide a radiotracer suitable for clinical imaging.

## 2. Experimental Part

### 2.1. Materials and Methods

Millipore water (Milli Q^®^ 8/16 Direct System, Merck, Rahway, NJ, USA), acetonitrile (ACN, for HPLC, gradient grade ≥ 99%, Sigma-Aldrich, Saint Louis, MO, USA) and trifluoroacetic acid (TFA, Sigma Aldrich) were used for HPLC. Ruthenium(III) chloride hydrate (Premion^®^, 99.99%, Alfa Aesar, Ward Hill, MA, USA and Johnson Matthey, London, UK), hydrochloric acid (HCl; 37%, VWR and 30% Suprapur^®^, Merck), acetonitrile (ACN; VWR), ethanol (absolute, EMPROVE^®^ exp Ph Eur, BP, JP, USP, Merck), methyl *tert*-butyl ether (MTBE; anhydrous 99.8%, Sigma Aldrich), indazole (Polivalent-95), cesium chloride (99+% pure, Acros Organics, Antwerp, Belgium), sodium sulfate (99% extra pure, anhydrous, Acros Organics), aluminum sulfate octadecahydrate (98+%, extra pure, Acros Organics), dimethyl sulfoxide (DMSO; anhydrous ≥99%, Sigma Aldrich) and 3-(4,5-dimethyl-2-thiazolyl)-2,5-diphenyl-*2H*-tetrazolium bromide (MTT; ≥97.5%, Sigma Aldrich) were used as supplied. All solutions of chemicals were prepared shortly before use.

Gamma-ray spectrometry measurements were performed using a high-purity germanium (HPGe) detector (Canberra, France). Full energy peak (FEP) efficiency calibration of the spectrometry was determined using a certified point-like source (provided by Physikalisch-Technische Bundesansalt (PTB)). The spectra were analyzed with Canberra’s Genie2000 software package version 3.3.

HPLC analysis was performed with a Chromolith^®^ performance RP-18e 100_4.6 mm column (Merck) on a Merck Hitachi L-6200A intelligent pump with a Merck Hitachi UV detector L-7400 and a Packard Radiomatic Flo-One Beta detector equipped with a BGO cell for radioactivity detection. For analysis, a gradient was applied using acetonitrile with 0.1% TFA (solvent A) and Milli Q water with 0.1% TFA (solvent B): 0–3 min, 10% A; 3–14 min, 55% A; 14–16 min, 95% A; 16–19.5 min, 95% A; 19.5–21 min, 10% A; 21–22 min, 10% A (A + B = 100%, flowrate = 3 mL/min, total run time = 22 min, UV wavelength = 220 nm). All samples were filtered through a 0.22 µm (Millex^®^-GV) filter before injection.

UV/Vis spectra were recorded on a Hitachi U-2000 Dual Beam spectrophotometer. Sample activities were measured on an ISOMED 2010 dose calibrator. For centrifugation during synthesis, a Fisherbrand^TM^ Mini-Centrifuge from Fisher Scientific was used. Radioactive organ samples were measured on Perkin Elmer Wizard^2^ gamma counter. Non-radioactive organ samples were prepared, and the ruthenium content was quantified with ICP-MS. The exact procedure including instrument information can be found in the Appendix A.

### 2.2. [^103^Ru]RuCl_3_·xH_2_O Production

[^103^Ru]RuCl_3_·xH_2_O was obtained by neutron activation of natural RuCl_3_·xH_2_O with the Production Neutron Activation (PNA) installation at the spallation neutron source SINQ at Paul Scherrer Institute [31] (Villigen, Switzerland). Quartz ampoules (OD = 8 mm, ID = 5 mm) were prepared by inserting 40–50 mg natural RuCl_3_·xH_2_O and sealing them, producing an ampoule with a length of 45–50 mm. They were, subsequently, irradiated for three weeks at a neutron flux φ = 4·10^13^ n cm^−2^ s^−1^. After irradiation, the ampoules were left to “cool” for one week before they were crushed and the target material was dissolved in 3 mL warm (T = 60 °C) 30% HCl. Radionuclide purity was assessed by means of γ-spectrometry.

### 2.3. Syntheses

KP1019 (indazolium *trans*-[tetrachloridobis(indazole)ruthenate(III)]) (**1a**): RuCl_3_·xH_2_O (50 mg, 0.24 mmol, 1 eq.) was dissolved in ethanol (0.3 mL) and conc. HCl (37%, 0.3 mL), and refluxed for 3 h. Indazole (189 mg, 1.60 mmol, 6.64 eq.) was dissolved in water (0.4 mL) and conc. HCl (37%, 4 mL), and then added to the solution and the mixture was refluxed for 3 h. After cooling to room temperature (RT), the formed solid was collected by filtration, washed twice with 2 M HCl and dried in vacuo. Yield: 128 mg (89%), brown solid.

Elemental analysis calcd (%). for C_21_H_19_Cl_4_N_6_Ru·0.25 H_2_O: C 41.84, H 3.26, N 13.94; found: C 41.77, H 2.95, N 13.83.

HPLC: *t*_R_ (Hind^+^) = 6.15 min, *t*_R_ ([Ru(ind)_2_Cl_4_]^−^) = 8.49 min.

Cesium *trans*-[tetrachloridobis(indazole)ruthenate(III)] (**2a**): **1a** (128 mg, 0.21 mmol) and CsCl (144 mg, 0.67 mmol, 2.8 eq.) were stirred in ethanol (2 mL) at RT for 2 h. The orange suspension was filtered and washed twice with ethanol. Then, the intermediate was stirred in a 2:1 *v*/*v* ethanol/water mixture at RT for 15 min. The solid was collected by filtration (using the same funnel as used previously), washed twice with ethanol and dried in vacuo. Yield: 116 mg (78%), red-brown solid.

Elemental analysis calcd (%). for C_14_H_12_Cl_4_N_4_CsRu: C 27.47, H 1.98, N 9.15; found: C 27.41, H 2.04, N 9.07.

HPLC: *t*_R_ ([Ru(ind)_2_Cl_4_]^−^) = 8.08 min.

BOLD-100 (sodium *trans*-[tetrachloridobis(indazole)ruthenate(III)]) (**3a**): Cesium salt **2a** (116 mg, 0.19 mmol) was suspended in a 1.1 M NaAl(SO_4_)_2_·18 H_2_O solution (1.4 mL). CsCl (spatula tip, catalytic amount) was added, and the mixture was stirred at RT for 24 h. The brown slurry was washed with saturated aqueous Na_2_SO_4_, until the filtrates were colorless (four times). Afterward, the residue was stirred in ACN (0.7 mL) at RT for 15 min. The insoluble sulfate salts were removed by filtration and the salt cake was rinsed with ACN until the filtrate was colorless (three times). The collected filtrates were diluted with MTBE (4:1) and set aside for 30 min. The precipitated product was filtered, washed twice with MTBE and dried in vacuo. Yield: 65 mg (54%), fluffy dark brown solid.

Elemental analysis calcd (%). for C_14_H_12_Cl_4_N_4_NaRu·1 H_2_O: C 32.33, H 2.71, N 10.77, O 3.08; found: C 32.38, H 2.72, N 10.86, O 3.30.

HPLC: *t*_R_ ([Ru(ind)_2_Cl_4_]^−^) = 8.23 min.

[^103^Ru]KP1019 ([^103^Ru]indazolium *trans*-[tetrachloridobis(indazole)ruthenate(III)]) (**1b**): Ethanol (1 mL) was added to 1.5 mL of the [^103^Ru]RuCl_3_·xH_2_O solution (16 mg, 0.077 mmol, 46.27 MBq, 1 eq. in 1.5 mL HCl, 37%) and the mixture was refluxed for 3 h. Indazole (61 mg, 0.51 mmol, 6.64 eq.) dissolved in H_2_O (0.1 mL) and conc. HCl (37%, 0.4 mL) was added to the ^103^RuCl_3_ solution. The reaction mixture was refluxed for 3 h. After cooling down to RT, the formed suspension was centrifuged, and the supernatant was removed. The remaining solid was washed with 2 M HCl (3 × 1 mL) and the obtained product (37 mg, 81%, 37.56 MBq, yield determined by isolated radioactivity) was analyzed using HPLC.

HPLC: *t*_R_ (Hind^+^) = 5.56 min, *t*_R_ ([^103^Ru][Ru(ind)_2_Cl_4_]^−^) = 8.16 min (UV), 8.61 min (radiochannel);

RCY: >76%;

RCP: >94%.

[^103^Ru]Cesium *trans*-[tetrachloridobis(indazole)ruthenate(III)] (**2b**): Indazolium salt **1b** (37 mg, 0.062 mmol) was suspended in ethanol (0.75 mL), CsCl (36 mg, 0.22 mmol, 2.8 eq.) and the mixture was stirred at RT for 2 h. The solid was separated by centrifugation, washed twice with ethanol (0.75 mL), and stirred in a 2:1 *v*/*v* ethanol water mixture (0.75 mL) for 15 min at RT. The product (32 mg, 67%, 30.67 MBq) was centrifuged, washed twice with ethanol (0.5 mL) and analyzed using HPLC.

HPLC: *t*_R_ ([^103^Ru][Ru(ind)_2_Cl_4_]^−^) = 7.96 min (UV), 8.38 min (radiochannel);

RCY: >62%;

RCP: >93%.

[^103^Ru]BOLD-100 ([^103^Ru]sodium *trans*-[tetrachloridobis(indazole)ruthenate(III)]) (**3b**): Cesium salt **2b** (32 mg, 0.052 mmol) was suspended in a 1.1 M NaAl(SO_4_)_2_·18 H_2_O solution. CsCl (spatula tip, catalytic amount) was added, and the mixture was stirred at RT for 24 h. The crude product was obtained by centrifugation and washing with a saturated Na_2_SO_4_ solution. The obtained solid was stirred in ACN (0.75 mL) for 15 min at RT. The suspension was centrifuged and washed with ACN (2 × 0.75 mL). The filtrates were combined, diluted with MTBE (5:1) and set aside for 1 h. The precipitated brown needles were collected by centrifugation, washed with MTBE and dried, yielding pure [^103^Ru]BOLD-100 (15.6 mg, 40%, 17.73 MBq).

HPLC: *t*_R_ ([^103^Ru][Ru(ind)_2_Cl_4_]^−^) = 8.16 min (UV), 8.65 min (radiochannel);

RCY: >38%;

RCP: >95%.

All UV/Vis spectra and HPLC chromatograms can be found in the Appendix A.

### 2.4. Cell Culture and Cytotoxicity Assays

The murine cancer cell line CT26 and the human cancer cell line HCT116 were both obtained from the American Type Culture Collection (ATCC). Both cell lines were grown as adherent monolayer cultures in 75 cm^2^ culture flasks in RPMI 1640 medium (supplemented with 2 mM l-glutamine and 1% PenStrep, Gibco^TM^) and 10% fetal bovine serum (FBS, Gibco^TM^) at 37 °C under a humid atmosphere containing 5% CO_2_ and 95% air. For splitting, accutase^®^ solution (Sigma Aldrich) was used.

Cell viability was determined by the colorimetric MTT assay. The cells were seeded in RPMI 1640 medium in a 96-well cell culture plate 24 h prior to treatment at a density of 1.5 × 10^3^ (CT26) and 2 × 10^3^ (HCT116) cells per well. A stock solution of compounds **3a** and **3b** was prepared in DMSO and then diluted with RPMI 1640 medium (maximum 0.5% *v*/*v* of DMSO per well). After serial dilutions, a 100 µL aliquot was added to each well (800, 400, 200, 100, 50, 25, 15.5, 6.25, 3.13 µmol/L per well). After incubation for 96 h, the drug solutions were replaced with 100 µL medium/MTT mixtures [6 parts of RPMI 1640 medium supplemented with 10% FBS, 2 mM L-glutamine and 1% PenStrep; 1 part of MTT solution in phosphate-buffered saline (5 mg/mL)]. After 4 h of incubation, the medium mixtures were replaced with 150 µL DMSO per well to dissolve the formed formazan crystals. Optical densities were measured at 490 nm with a Synergy^TM^ HTX Multi-Mode Reader (BioTek, Winooski, VT, USA), using a reference wavelength of 650 nm to correct for unspecific absorption. Concentration–effect curves were calculated relative to untreated controls. Half maximal inhibitory concentration (IC_50_) values were interpolated with Gen5^TM^ software version 2.09.

### 2.5. Animals

Eight- to twelve-week-old female Balb/c mice were purchased from Envigo (San Pietro al Natisone, Italy). The animals were kept in a pathogen-free environment and every procedure was performed in a laminar airflow cabinet. The experiments were performed according to the regulations of the Ethics Committee for the Care and Use of Laboratory Animals at the Medical University of Vienna (approval number BMWF-66.009/0394-V/3b/2018), the U.S. Public Health Service Policy on Human Care and Use of Laboratory Animals as well as the United Kingdom Coordinating Committee on Cancer Prevention Research’s Guidelines for the Welfare of Animals in Experimental Neoplasia.

For the evaluation of organ distribution, CT26 cells (5 × 10^5^ in serum-free medium) were injected subcutaneously into the right flank of Balb/c mice. Animals were controlled for distress development every day and tumor size was assessed regularly using caliper measurements. CT26 cells were inoculated 10 days prior to drug application. Tumor volume was calculated using the formula: (length × width^2^); the tumors reached a size of about 500 mm^3^. Animals were treated once intravenously (i.v.) via tail-vein injection with **3a** or **3b** at a dose of 30 mg/kg (dissolved in 0.9% NaCl, 0.6 mg/100 µL/20 g mouse) and a specific activity of 0.64–1.36 MBq/mg for compound **3b**. The mice were anesthetized and sacrificed by cervical fracture 4, 24, 48 and 72 h p.i., with *n* ≥ 2 (**3a**) and *n* ≥ 3 (**3b**) for each time point. Samples of tumor, liver, kidney, muscle, heart, lung, spleen, pancreas, stomach, colon, small intestine, brain, bone, and blood were collected from each mouse. After clotting at RT, serum was isolated from the collected blood samples by centrifugation at 3000 rpm for 10 min twice and stored together with the collected tissue samples at −20 °C. Finally, the ruthenium content was quantified by ICP-MS in the case of the non-radioactive BOLD-100 (**3a**) and via gamma counting for [^103^Ru]BOLD-100 (**3b**).

### 2.6. Statistics

Biodistribution data of Ru complexes were expressed as the average ± standard deviation and compared using two-way ANOVA with Sidak’s correction (Graph Pad Prism version 8.0.2). Values of *p* < 0.05 were considered statistically significant (* *p* ≤ 0.05; ** *p* ≤ 0.01; *** *p* ≤ 0.001).

## 3. Results and Discussion

### 3.1. Production of ^103^RuCl_3_

[^103^Ru]RuCl_3_·xH_2_O was obtained by irradiation of ^nat^RuCl_3_·xH_2_O with a neutron flux over a period of three weeks. Five ampoules containing 40–50 mg RuCl_3_·xH_2_O and activities up to 185 MBq were obtained (3.7–4.7 MBq/mg). After irradiation, the ampoules were crushed, and the target material was dissolved in 3 mL concentrated hydrochloric acid (HCl). Dissolution in warm conc. HCl dramatically increased the final activity of the isolated ^103^Ru samples. In addition to ^103^Ru, ^97^Ru (t_1/2_ = 2.9 d) and trace amounts of ^192^Ir (t_1/2_ = 73.83 d) activities in the samples were also determined. Representative gamma-ray spectrum of [^103^Ru]RuCl_3_ is provided in the Appendix A.

### 3.2. Synthesis

The three-step synthesis (Figure 1) of non-radioactive BOLD-100 was published and patented in 2018. Some experimental procedures of the synthesis were modified to allow the adaption of radiochemical synthesis (e.g., centrifugation instead of filtration) [32]. ^nat^RuCl_3_ was employed for the synthesis of non-radioactive BOLD-100, whereas c.a. [^103^Ru]RuCl_3_ (1.8–4.2 MBq/mg) was used for the preparation of [^103^Ru]BOLD-100. First, ^nat/103^RuCl_3_ was refluxed in HCl (37%) and ethanol for 3 h to ensure the absence of Ru(IV) species [33] and then reacted with indazole at elevated temperatures. The obtained indazolium complexes **1a** and **1b** were converted at RT to the corresponding cesium salts **2a** and **2b** upon treatment with an excess of cesium chloride. The cation exchange from cesium to sodium was performed in an aqueous solution of sodium aluminum sulfate. Pure products **3a** and **3b** were obtained by precipitation from acetonitrile/methyl *tert*-butyl ether. The yields, based on the initial amount of ^nat/103^RuCl_3_, were good, ranging from 89% (**1a**) and 81% (**1b**) for the indazole salt, to 78% (**2a**) and 67% (**2b**) for the cesium salt and 54% (**3a**) and 38% (**3b**) for the final products. All complexes were characterized by RP-HPLC chromatography and UV/Vis spectroscopy (Appendix A). In addition, unlabeled compounds **1a**–**3a** were characterized by elemental analysis. The radiochemical purity for compounds **1b**–**3b** was >93% and the overall radiochemical yield for **3b** was >38%.

### 3.3. Cytotoxicity In Vitro

The antiproliferative activities of compounds **3a** and **3b** were determined in human colon carcinoma (HCT116) and murine colon carcinoma (CT26) cell lines using the colorimetric MTT assay with an exposure time of 96 h. This assay was chosen to compare the cytotoxic profile of [^103^Ru]BOLD-100 with that of the non-radioactive drug BOLD-100. These cell lines were selected because BOLD-100 showed promising cytotoxic activities against colon carcinoma cell lines, like HCT116^11^, and CT26 was chosen because it was used for the in vivo experiments. The respective cell survival curves can be found in the Appendix A. The IC_50_ values for both compounds in the utilized cell lines showed no significant differences (Table 2), which could be explained by the low specific activities of **3b** (0.5–1.4 MBq/mg). This result shows that the radiolabeling of BOLD-100 did not affect the biological activity in vitro. In order to study the potential combination of therapeutic effects, experiments with [^103^Ru]BOLD-100 with a higher specific activity would be needed.

### 3.4. Biodistribution Experiments

The tissue distributions of compounds **3a** and **3b** were studied in Balb/c mice bearing CT26 allografts over a time period of 72 h. In this experiment, we aimed to evaluate whether there are any differences regarding the biodistribution of the compounds. In addition, we sought to identify an appropriate time point with a suitable tumor-to-background ratio for future imaging experiments using ^97^Ru. The time period of 72 h was chosen to match the physical half-life of ^97^Ru (t_1/2_ = 2.8 days). Figure 2 shows the Ru levels of the selected organs. The data for all time points and the calculated tumor-to-organ ratios can be found in the Appendix A.

In general, both isotopomers of BOLD-100 had an almost identical biodistribution profile. Only the kidneys and the liver showed a statistically significant difference at some time points and therefore, the following discussion relates to both compounds. The highest uptake of ruthenium at 4 h was found in the blood, which is in good agreement with the proposed mode of action of BOLD-100, which involves the binding of the drug to blood proteins such as albumin in the first step [10]. In accordance with this, the amount of blood serum-bound Ru species was highest at 4 h p.i. and decreased slowly over time. Relatively high Ru levels were measured in the lungs, presumably caused by the high blood flow to this organ. The observed elevated uptake of ruthenium in the liver and kidneys is likely related to their role as excretion and clearance organs. The lowest amounts of Ru were found in the bones, muscles, and brain. The Ru uptake was also rather low in all other investigated organs and tissues (Appendix A), generally resulting in good tumor-to-background ratios, especially at later time points (Appendix A). Previously published data on the biodistribution of BOLD-100 in Balb/c mice over 24 h are in good agreement with our results within this period of time, even though we injected a smaller amount of BOLD-100 [34]. The amounts of Ru decreased over time in all organs except the kidneys (Appendix A). In contrast, the tumor uptake steadily increased within the first day and the highest levels were found at 24 h p.i., with 8.4%ID/g (**3a**) and 6.4%ID/g (**3b**). Although there was no statistically significant difference in tumor tissues, the slightly lower tumor uptake of compound **3b** compared to **3a** at 24, 48 and 72 h, could be explained by the different measurement methods. While the entire tumor was measured for the radioactive tumor samples, only a small part of the allograft was used for the measurements of the non-radioactive tumors via ICP-MS. In the case of hypoxic or necrotic regions of the tumor, the drug uptake might therefore appear to be decreased. Provided that the samples were processed equally, a recent publication by Wallimann et al. showed that ICP-MS is a valid and alternative method for the characterization of metal conjugates [35]. The uptake of the Ru complexes in tumors was moderate; however, it was increased in comparison to most other organs and tissues (Appendix A) except for the liver and kidneys. We hypothesize that the biodistribution might be influenced by the amount of the compounds injected. Studies employing sub-equimolar doses of BOLD-100 relative to mouse serum albumin are currently being planned.

## 4. Conclusions

We report the first synthesis and characterization of radiolabeled c.a. [^103^Ru]BOLD-100. A direct comparison with the non-radioactive chemotherapeutic agent BOLD-100 in vitro and in vivo showed that there were no significant differences between the compounds in terms of cytotoxicity and pharmacokinetics. The result of the present study with c.a. [^103^Ru]BOLD-100 suggests that the analogue complex [^97^Ru]BOLD-100 may represent a promising SPECT imaging probe for the stratification of patients for a BOLD-100 therapy. Studies with [^97^Ru]BOLD-100 including imaging studies are currently ongoing and will be reported in due time.

## Data Availability

Data are contained within the article and Appendix A.

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
