# Peer review of "Synthesis and Preclinical Evaluation of Radiolabeled [103Ru]BOLD-100"

_pharmaceutics, 2023, doi:10.3390/pharmaceutics15112626_

Round 1
Reviewer 1 Report
Comments and Suggestions for Authors
The manuscript entitled Synthesis and preclinical evaluation of radiolabeled [103Ru]BOLD-100 is focused on the preparation of a radiolabeled compound derived from a metallodrug and its further comparison with the non-labelled parent antitumor compound.
Despite the manuscript is well presented and offers relevant preclinical assays, the manuscript contains some minor issues that must be clarified prior to publication.
Along the manuscript there could be found several missing citations (Error! Reference source not found) that must be removed from the text or be properly cited.
Additionally, relevant characterization information is missing all along the manuscript. All compounds, either radioactive or not are not properly analyzed neither referenced adequately. For instance, in the synthesis of both BOLD100 compounds, no clue is given about the possible coexistence of cis compounds along with the claimed trans ones. Moreover in the synthesis of compound KP1019, HPLC analysis seems to be a mixture but microanalysis coincides with the exact composition of desired compound. Please, clarify this and include relevant NMR analysis to relevant compounds described and compare with published data.
Please clarify the purpose of using eight- to seventeen-week-old female Balb/c mice. Is there a non-discussed purpose on using animals of different ages?
Author Response
Dear Editor
Please see uploaded pdf file for our reply to the comments of reviewer 1.
Kind regards,
Tom Mindt

Reviewer 2 Report
Comments and Suggestions for Authors
Please see the attached file.

Author Response
Dear Editor
Please see uploaded pdf file for our reply to the comments of reviewer 2.
Kind regards,
Tom Mindt

Reviewer 3 Report
Comments and Suggestions for Authors
Dr. Thomas L. Mindt etc. reported a radiolabeled version BOLD-100, [103Ru] BOLD-100, was synthesized and characterized. Both the original and radiolabeled versions were tested in vitro and in vivo. The study found that the cytotoxicity and pharmacokinetic profiles of BOLD-100 and [103Ru] BOLD-100 were identical.
The comparison between radioactive [103Ru] BOLD-100 and non-radioactive chemotherapeutic agent BOLD-100 did not show meaningful depending on the context and objectives of the study.
Here are the considerations:
1. Mechanism of Action: If the radioactive [103Ru] BOLD-100 works via a different mechanism compared to the non-radioactive version, then comparing their efficacy, safety, or other relevant parameters could provide important insights. In the present manuscript, this specific comparison was not addressed.
2. Synergistic Effects: There could be a synergistic effect where the radioactive decay of [103Ru] could enhance the therapeutic efficacy of BOLD-100. Conversely, it's also possible that the radioactivity could interfere with the drug's mechanism, rendering it less effective. There was not showing these synergistic effects in the current manuscript.
3. Safety and Toxicity: The addition of a radioactive isotope can change the safety profile of a compound. A comparison would be meaningful to determine if the radioactive variant has a higher or different toxicity profile. However, the authors reported no significant differences between the compounds in terms of cytotoxicity and pharmacokinetics.
4. Drug Delivery and Biodistribution: Radioactive isotopes can be used as tracers to study the biodistribution of a compound. The manuscript was comparing how each version of BOLD-100 distributes in the body, this was a valuable contribution.
5. Specificity of the Study: The aim was just to observe the basic difference between a radioactive and a non-radioactive substance without providing a clear rationale or without connecting it to broader therapeutic implications, the comparison seems not meaningful.
6. Assessment Parameters: The parameters assessed in the study will also determine the meaningfulness. However, the study only looked at basic parameters without a clear clinical implication, it was of lesser value compared to a study that evaluates therapeutic efficacy, side effects, or other clinically relevant outcomes.
In summary, the report did not show meaningful, depends on the context, objectives, and outcomes of the study.
Comments on the Quality of English LanguageThe manuscript is well-written, and the quality of the English language is commendable.
Author Response
Dear Editor
Please see uploaded pdf file for our reply to the comments of reviewer 3.
Kind regards,
Tom Mindt
